

# Differentiation of *Bifidobacterium longum* subspecies *longum* and *infantis* by quantitative PCR using functional gene targets

Blair Lawley[1], Karen Munro[1], Alan Hughes[1], Alison J. Hodgkinson[2], Colin G. Prosser[3], Dianne Lowry[3], Shao J. Zhou[4,5], Maria Makrides[6], Robert A. Gibson[5], Christophe Lay[7], Charmaine Chew[7], Pheng Soon Lee[8], Khai Hong Wong[8] and Gerald W. Tannock[1,9,10]

[1] Department of Microbiology and Immunology, University of Otago, Dunedin, New Zealand
[2] AgResearch, Ruakura, Hamilton, New Zealand
[3] Dairy Goat Cooperative (NZ) Ltd., Hamilton, New Zealand
[4] Women's and Children's Health Research Institute, Adelaide, Australia
[5] School of Agriculture, Food and Wine, University of Adelaide, Adelaide, Australia
[6] Healthy Mothers, Babies and Children, South Australian Health and Medical Research Institute, Adelaide, Australia
[7] Danone Nutricia Research, Biopolis, Singapore
[8] Mead Johnson Nutrition, Marina Bay, Singapore
[9] Riddet Institute Centre of Research Excellence, Palmerston North, New Zealand
[10] Microbiome Otago, University of Otago, Dunedin, New Zealand

Corresponding author
Blair Lawley, blair.lawley@otago.ac.nz

## ABSTRACT

**Background**. Members of the genus *Bifidobacterium* are abundant in the feces of babies during the exclusively-milk-diet period of life. *Bifidobacterium longum* is reported to be a common member of the infant fecal microbiota. However, *B. longum* is composed of three subspecies, two of which are represented in the bowel microbiota (*B. longum* subsp. *longum*; *B. longum* subsp. *infantis*). *B. longum* subspecies are not differentiated in many studies, so that their prevalence and relative abundances are not accurately known. This may largely be due to difficulty in assigning subspecies identity using DNA sequences of *16S rRNA* or *tuf* genes that are commonly used in bacterial taxonomy.
**Methods**. We developed a qPCR method targeting the sialidase gene (subsp. *infantis*) and sugar kinase gene (subsp. *longum*) to differentiate the subspecies using specific primers and probes. Specificity of the primers/probes was tested by *in silico,* pangenomic search, and using DNA from standard cultures of bifidobacterial species. The utility of the method was further examined using DNA from feces that had been collected from infants inhabiting various geographical regions.
**Results**. A pangenomic search of the NCBI genomic database showed that the PCR primers/probes targeted only the respective genes of the two subspecies. The primers/probes showed total specificity when tested against DNA extracted from the gold standard strains (type cultures) of bifidobacterial species detected in infant feces. Use of the qPCR method with DNA extracted from the feces of infants of different ages, delivery method and nutrition, showed that subsp. *infantis* was detectable (0–32.4% prevalence) in the feces of Australian ($n = 90$), South-East Asian ($n = 24$), and Chinese

babies ($n = 91$), but in all cases at low abundance (<0.01–4.6%) compared to subsp. *longum* (0.1–33.7% abundance; 21.4–100% prevalence).

**Discussion.** Our qPCR method differentiates *B. longum* subspecies *longum* and *infantis* using characteristic functional genes. It can be used as an identification aid for isolates of bifidobacteria, as well as in determining prevalence and abundance of the subspecies in feces. The method should thus be useful in ecological studies of the infant gut microbiota during early life where an understanding of the ecology of bifidobacterial species may be important in developing interventions to promote infant health.

## INTRODUCTION

Bifidobacteria commonly dominate the fecal microbiota of infants during the exclusively milk-fed period of life (*Biavati et al., 1984*; *Favier et al., 2002*; *Young et al., 2004*; *Roger et al., 2010*; *Grönlund et al., 2011*; *Tannock et al., 2013*; *Turroni et al., 2012*; *Makino et al., 2013*; *Bäckhed et al., 2015*; *Milani et al., 2015*; *Martin et al., 2016*). This observation can be at least partly explained by the fact that bifidobacterial species that are enriched in the infant bowel can utilize Human Milk Oligosaccharides (HMO) or their components as growth substrates (*Garrido, Dallas & Mills, 2013*). *B. longum* is commonly detected in studies of the infant fecal microbiota. However, *B. longum* is composed of three subspecies, two of which are represented in the bowel microbiota (*B. longum* subsp. *longum*; *B. longum* subsp. *infantis*). Differentiation of *B. longum* into subspecies has been carried out in relatively few quantitative studies of the infant fecal microbiota, so that subspecies prevalence and relative abundances are not accurately known on a global scale (*Grönlund et al., 2011*; *Huda et al., 2014*; *Martin et al., 2016*). This is probably due to the difficulty of differentiating the *B. longum* subspecies using *16S rRNA* gene sequences (*Youn, Seo & Ji, 2008*). Previously, we tested primers targeting the *16S rRNA* gene and found weak cross-reactivity between subspecies when high concentrations of non-target template were present (*Tannock et al., 2013*). Other groups have used *16S rRNA* gene primers (*Matsuki et al., 1999*), 16S/23S intergenic region primers (*Haarman & Knol, 2006*), *23S rRNA* gene primers (*Kurakawa et al., 2015*) and *tuf* gene primers (*Sheu et al., 2010*) to differentiate the subspecies, but a high degree of similarity between sequences of marker genes, such as the *16S rRNA* gene and *tuf* gene, leads to difficulty in generating PCR primers that are capable of absolute discrimination of the *B. longum* subspecies (*Youn, Seo & Ji, 2008*). Inaccuracies in taxonomic assignment (*O'Callaghan et al., 2015*) may also contribute to the difficulty in designing discriminatory primers and/or probes based on all available sequence data. We believe an approach targeting genes that define the functional differences between *B. longum* subspecies, rather than phylogenetic marker genes, may be more appropriate and describe here a qPCR method that distinguishes subsp. *longum* from subsp. *infantis* by targeting functional genes exclusive to each of these subspecies. The method was used to determine the prevalence and relative abundances of

these two subspecies in the feces of children in Australia, South-East Asia, and China that were breast milk or formula-fed, or delivered vaginally or by caesarean section.

## MATERIALS AND METHODS

### Quantitative PCR differentiation of *B. longum* subsp. *longum* and subsp. *infantis*

Primers and Taqman® probes were designed to target the *B. longum* subsp. *infantis* sialidase gene (locus tag 'Blon2348' from strain ATCC 15697, NCBI Reference Sequence: NC_011593.1) and the *B. longum* subsp. *longum* sugar kinase gene (locus tag 'BL0274' from strain NCC 2705, NCBI Reference Sequence: NC_004307.2) using 'primer 3' (*Untergasser et al., 2012*). Primers and probes were obtained from IDT (Singapore) and are described in Table 1. The primer/probe combinations were tested for reaction efficiency and specificity using genomic DNA (gDNA) purified from bifidobacterial type cultures (the gold standard cultures for species) of species reported to be detected in infant feces: *B. adolescentis* (DSM 20083[T]), *B. animalis* subsp. *lactis* (DSM 10140[T]), *B. angulatum* DSM 20098[T], *B. bifidum* (DSM 20456[T]), *B. breve* (ATCC 15700[T]), *B. catenulatum* (DSM 20224[T]), *B. dentium* (ATCC 27534[T]), *B. longum* subsp. *infantis* (DSM 20088[T]), *B. longum* subsp. *longum* (ATCC 15707[T]), *B. pseudocatenulatum* (DSM 20438[T]), and *B. pseudolongum* (ATCC 25526[T]), (*Grönlund et al., 2011*; *Makino et al., 2013*; *Huda et al., 2014*; *Bäckhed et al., 2015*; *Vazquez-Gutierrez et al., 2015*; *Martin et al., 2016*). A Life Technologies ViiA™7 real time PCR system and MicroAmp Fast optical 96-well or 384-well plates with optical adhesive film (Applied Biosystems, Carlsbad, CA, USA) were used. All reactions were carried out in a final volume of 15 μl containing 1 × TaqMan® Fast PCR mastermix (Applied Biosystems), 300 nM of each primer and 100 nM TaqMan® probe. For specificity testing, template DNA was diluted to 5 ng/μl, and 2 ng was added to each reaction. The thermocycling profile consisted of an initial activation of the polymerase at 95 °C for 30 s, followed by 40 cycles of 95 °C for 10 s and 60 °C for 30 s. Fluorescence levels were measured after the 60 °C annealing/extension step. Standard curves (to measure reaction efficiency) were generated using gDNA extracted from bifidobacterial strains *Bifidobacterium longum* subsp. *longum* (ATCC 15707[T]) and *Bifidobacterium longum* subsp. *infantis* (DSM 20088[T]) using the bead-beating phenol/chloroform/ethanol protocol described previously (*Tannock et al., 2013*). The standard DNA was quantified spectrophotometrically using a NanoDrop 1,000 spectrophotometer (Thermo Scientific, Waltham, MA, USA) and diluted in 10-fold steps from $5 \times 10^6$ to $5 \times 10^1$ genomes/reaction, calculated using target gene copies per genome obtained from genome sequence information (NCBI). All reactions were carried out in duplicate and were run twice on separate plates. No-template controls were also included on each plate. Reactions in which duplicate Ct values varied by more than 0.5 Ct's were also repeated.

### Determination of the prevalence and abundances of subspecies *longum* and *infantis* in infant feces

Standard curve-based qPCR was used to determine the relative abundance and prevalence of *B. longum* subspecies. The CFU/ml of *B. longum* subsp. *longum* (ATCC 15707[T]) and
**Table 1  PCR primers and probes.**

| Target | Primer/Probe | Sequence 5′–3′ | Reference |
|---|---|---|---|
| *B. longum* subsp. *infantis* sialidase gene | inf_2348_F | ATACAGCAGAACCTTGGCCT | This study |
| | inf_2348_R | GCGATCACATGGACGAGAAC | |
| | inf_2348_P | /FAM/TTTCACGGA/ZEN/TCACCGG ACCATACG/3IABkFQ/ | |
| *B. longum* subsp. *longum* sugar kinase gene | lon_0274_F | GAGGCGATGGTCTGGAAGTT | This study |
| | lon_0274_R | CCACATCGCCGAGAAGATTC | |
| | lon_0274_P | /56-FAM/AATTCGATG/ZEN/CCCAGCG TGGTCTT/3IABkFQ/ | |
| All bacteria *16S rRNA* gene | Uni_F | ACTCCTACGGGAGGCAGCAGT | *Wang, Cao & Cerniglia (1996)* |
| | Uni_R | ATTACCGCGGCTGCTGGC | |

*B. longum* subsp. *infantis* (DSM 20088[T]) was determined by spreading dilutions of 7-h anaerobic cultures (Lactobacilli MRS broth; Difco, Leeuwarden, The Netherlands) on Lactobacilli MRS agar to obtain colony counts after 48 h incubation at 37 °C in an anaerobic glovebox. Genomic DNA was extracted from 1 ml of culture using the bead-beating phenol/chloroform/ethanol protocol described previously (*Tannock et al., 2013*). Standard curves were generated with template quantities equivalent to $5 \times 10^6 – 5 \times 10^1$ CFU in a 10-fold dilution series. Total community *16S rRNA* gene targets were determined using universal primers and a SYBR® green detection system as previously described (*Tannock et al., 2013*) whilst *B. longum* subspecies target numbers were obtained using the subspecies-specific Taqman® assays. Relative abundance of each subspecies was determined by dividing the subspecies target quantity by the total community target quantity and multiplying by 100. If a *B. longum* subspecies assay was positive, the sample was assumed to be positive for this subspecies. Prevalence was determined by dividing the number of fecal samples positive for each subspecies by the total number of samples per infant group, and multiplying by 100.

## DNA extraction from infant feces

The methodology for DNA extraction from feces of Australian and Chinese children was described by *Tannock et al. (2013)* and included bead-beating to disrupt bacterial cells, phenol-chloroform treatment, and ethanol precipitation. South-East Asian fecal DNA was extracted from feces using Qiagen DNA Stool Mini-Kit (Qiagen, Hilden, Germany) modified with bead-beating. In brief, approximately 200 mg of 0.1 mm glass beads were added to 200 mg of stool sample and suspended in QIAGEN ASL buffer. Bead-beating used a FastPrep-24 (M.P. Biomedicals, Santa Ana, CA, USA) for 3 repetitions of 1 min bead-beating with 5 min incubation on ice. Samples were then heated at 95 °C for 15 min before centrifugation at 20,000× g for 1 min. The supernatant was transferred to a clean tube containing an InhibitEX tablet and vortexed to mix. The manufacturer's instructions were followed thereafter. Extracted fecal DNA were eluted in 50 μl AE buffer, checked on NanoDrop 2000 and stored at −20 °C prior to analysis.

## Effect of DNA extraction method on detection of *B. longum* subspecies

As two distinct gDNA extraction methods were used in this study, we determined whether the choice of extraction methodology would significantly impact on the quantitative detection of *B. longum* subspecies. We extracted gDNA from one Australian and three Chinese infant fecal samples using the original bead-beating/phenol-chloroform/ethanol precipitation method and two commercial silica membrane-based extraction methods. The two commercial methods were the Qiagen DNA Stool Mini-Kit (with modifications described above) and the MoBio PowerSoil® DNA isolation kit (MoBio, Carlsbad, CA, USA) which was used according to the manufacturer's instructions. Extracted gDNA was used as template in separate qPCR reactions including the universal *16S rRNA* gene primers (to determine total community target quantity) and each of the *B. longum* subspecies Taqman® assay primer/probe sets (to determine target quantity for each *B. longum* subspecies). Single aliquots of each fecal sample were extracted by each method and qPCR reactions were carried out in duplicate as described above.

### Australian infants

The Australian infants included in this study were part of a larger study (Australian New Zealand Clinical Trials Registry ACTRN12608000047392) (*Zhou et al., 2014*) in South Australia, comparing growth and nutritional status of infants fed either goat milk-based infant formula or cow milk-based infant formula (Dairy Goat Co-operative (NZ) Ltd., Hamilton, New Zealand). Healthy term infants were recruited to a multicenter, double-blind, controlled feeding trial. Infants were randomly allocated (stratified by sex and study center) to receive either goat milk or cow milk formula before they were two weeks of age. Infants were exclusively fed the study formulas (with no other liquids or solids except for water, vitamin or mineral supplements, or medicines). Groups of 30 infants were randomly selected from the main study for fecal microbiota analysis, and a parallel group of exclusively breast milk-fed infants was included as a reference group (30 infants). Fecal samples were obtained when the children were two months of age. Ethical approval for the work contained in this article was obtained from the WCHN Human Research Ethics Committee. Other details of the infants are given in *Tannock et al. (2013)*.

### South-East Asian infants

The infants (various ethnicities) were part of a larger clinical study registered under the Dutch Trial Register (NTR 2838). The infants were recruited as part of a multi-country, exploratory, randomized double-blind, controlled study in Singapore ($n = 19$) and Thailand ($n = 5$). An equal distribution of vaginally and elective caesarean section delivered infants were selected from each location. The infants were 'mixed fed' (breast-milk and/or cow milk-based infant formula) from birth. Fecal samples collected at Day 3/5 (hereafter termed < 1 week), two months and four months of age were selected for DNA extraction and qPCR analysis.

### Chinese infants

The Chinese infants included in this study were part of Mead Johnson Nutrition Clinical Protocol Number 8602. Healthy, singleton term infants, delivered by caesarian section, were recruited to the trial. Infants ($n = 40$) were exclusively fed formula (with no other liquids or solids except for water, vitamin or mineral supplements, or medicines). A parallel group of exclusively breast milk-fed infants was included as a reference group ($n = 51$). Fecal samples were obtained when the children were 6, 8, 10 and 12 weeks of age. Ethical approval for the work contained in this article was obtained from the Ethics Committee, Xin Hua Hospital, affiliated to the School of Medicine, Shanghai Jiao Tong University, China.

## RESULTS

### Primer/probe design, specificity and efficiency

The main aim of this study was to develop subspecies-specific quantitative PCR assays, focusing on functional attributes of *B. longum* subsp. *longum* and *infantis*. Target genes were chosen by submitting gene sequences within the HMO utilization region (*Sela et al., 2008*) for functional identification of *B. longum* subspecies *infantis* and gene sequences within the arabinose utilization region for functional identification of *B. longum* subsp. *longum* to BLAST searches against the NCBI nr database. Based on BLAST search results, genes present in both subspecies (or other bifidobacterial species) were rejected and optimal target genes were identified as the *B. longum* subsp. *infantis* sialidase gene (locus tag 'Blon2348' from strain ATCC15697) and the *B. longum* subsp. *longum* sugar kinase gene (locus tag 'BL0274' from strain NCC 2705).

Designed primers were checked for target specificity in-silico using Primer-BLAST (https://www.ncbi.nlm.nih.gov/tools/primer-blast/) against the NCBI nr database, and showed absence of predicted cross reactivity with non-target templates.

Reaction specificity and efficiency was tested by generating ten-fold serial dilutions of both *B. longum* subspecies *longum* and *B. longum* subspecies *infantis* gDNA so that reactions would contain between $5 \times 10^6$ and $5 \times 10^1$ genomes/reaction. For both of the primer/probe sets, amplification product was only obtained for the appropriate specific target, and not in the non-target template reactions (including those of other bifidobacterial species). Both primer/probe sets achieved reaction efficiencies between 88% and 92% across four separate runs. Modifications to annealing/extension temperatures, position of primers within the target gene and primer concentration did not lead to improvements in reaction efficiency. Reaction efficiency was also tested in the presence of a complex mixture of fecal DNA lacking bifidobacterial sequences. This DNA was obtained from the feces of an infant observed to be free of bifidobacterial sequences as determined by *16S rRNA* gene (v1-3) high-throughput sequencing. Addition of 1 ng/μl of this fecal genomic DNA did not impact on the efficiency of either primer/probe set. Further, reaction Ct values were identical in the presence or absence of this spiked fecal genomic DNA, confirming primer specificity in a complex background.
**Table 2  Effect of DNA extraction method.**

| Sample | Extraction method | B. longum subsp. longum[a] | B. longum subsp. infantis[a] |
|--------|-------------------|----------------------------|------------------------------|
| 58,550 | MoBio | 51.75 (4.61) | 0.00 (0.00) |
| 58,550 | Phenol | 34.07 (1.40) | 0.00 (0.00) |
| 58,550 | Qiagen | 66.85 (1.08) | 0.00 (0.00) |
| AF26B | MoBio | 1.66 (0.02) | 0.14 (0.01) |
| AF26B | Phenol | 0.18 (0.01) | 0.19 (0.00) |
| AF26B | Qiagen | 0.72 (0.08) | 0.62 (0.10) |
| AF12A | MoBio | 78.61 (1.91) | 0.00 (0.00) |
| AF12A | Phenol | 75.17 (2.26) | 0.00 (0.00) |
| AF12A | Qiagen | 75.77 (1.11) | 0.00 (0.00) |
| AF92A | MoBio | 0.20 (0.01) | 0.00 (0.00) |
| AF92A | Phenol | 0.02 (0.00) | 0.00 (0.00) |
| AF92A | Qiagen | 0.06 (0.00) | 0.00 (0.00) |

**Notes.**
[a]Mean % (SEM) of total microbiota determined by differential qPCR.

## Effect of differing genomic DNA extraction methods

Different DNA extraction methods were used in the infant study groups. Therefore, the potential effect of extraction method was determined. Genomic DNA was extracted from the feces of four infants using the phenol/chloroform/beadbeating method, the MoBio PowerSoil kit and the Qiagen DNA Stool Mini kit. Results indicated that extraction methodology would have a negligible impact on the detection of the *B. longum* subspecies (Table 2) but nevertheless argue for a standard extraction method to be used in all studies.

## Prevalence and relative abundances of subspecies *longum* and *infantis* in feces

*B. longum* subsp. *infantis* and subsp. *longum* were detected in the feces of infants located in three different geographical regions (Table 3). The prevalence of subspecies *infantis* tended to be less than that of subspecies *longum*, and was mainly influenced by the age of the child rather than nutrition; only two infants less than one week of age had detectable levels (<0.05% relative abundance) of subspecies *infantis*.

In general, subspecies *infantis* was present in low abundance in infant feces compared to *B. longum* subsp. *longum* (Table 4). The relative abundances of subspecies *infantis* were greatest in Chinese samples and lowest in Australian children. In the case of South-East Asian and Australian infants, caesarean section coupled with milk formula tended to be associated with low abundances of subspecies *infantis* (Table 4). In cow's milk formula-fed Chinese infants, the distribution of subspecies *longum* abundances tended to be different to those of breast milk-fed infants (Fig. 1). A similar differential distribution was not observed for subspecies *infantis* (Fig. S1). All of the Chinese infants had been delivered by caesarean, thus the subspecies *longum* abundances were affected by infant nutrition. Overall, it was concluded that subspecies *infantis* was prevalent, but was of low relative abundance compared to subspecies *longum* in infant feces collected in all three geographical regions.

Table 3 Prevalence of *B. longum* subsp. *longum* and subsp. *infantis* in the feces of infants as detected by qPCR.

| Geographical location | Nutrition | Delivery | Age at sampling (weeks) | *B. longum* subsp. *longum*[b] | *B. longum* subsp. *infantis*[b] |
|---|---|---|---|---|---|
| Chinese[a] | | | 6 | 86.2 | 24.1 |
| | Breast milk | Caesarean | 8 | 95.8 | 12.5 |
| | | | 10 | 100.0 | 11.1 |
| | | | 12 | 82.6 | 17.4 |
| | Cow's milk formula | Caesarean | 6 | 64.9 | 27.0 |
| | | | 8 | 62.2 | 32.4 |
| | | | 10 | 81.1 | 27.0 |
| | | | 12 | 78.4 | 32.4 |
| Australian[a] | Breast milk | Vaginal | 8 | 100.0 | 13.0 |
| | | Caesarean | 8 | 85.7 | 14.3 |
| | Cow's milk formula | Vaginal | 8 | 100.0 | 4.8 |
| | | Caesarean | 8 | 100.0 | 0.0 |
| | Goat's milk formula | Vaginal | 8 | 100.0 | 4.8 |
| | | Caesarean | 8 | 100.0 | 0.0 |
| South-East Asian[a] | Breast milk supplemented with cow's milk formula | Vaginal | <1 | 50.0 | 8.3 |
| | | | 8 | 58.3 | 8.3 |
| | | | 16 | 58.3 | 25.0 |
| | | Caesarean | <1 | 21.4 | 7.1 |
| | | | 8 | 53.9 | 15.4 |
| | | | 16 | 84.6 | 15.4 |

Notes.
  [a]Chinese babies, *n* = 51 caesarean delivery/breast milk, *n* = 40 caesarean delivery/formula; Australian babies, *n* = 30 breast milk (23 vaginal delivery, 7 caesarean), *n* = 30 cow's milk formula (21 vaginal delivery, 9 caesarean), *n* = 30 goat's milk formula (21 vaginal, 9 caesarean); South-East Asian babies, *n* = 12 vaginally delivery/breast milk and/or cow's milk formula, *n* = 12 caesarean delivery/breast milk and/or cow's milk formula.
  [b]% of infants harboring the subsp. as determined by differential qPCR.

# DISCUSSION

High-throughput sequencing of *16S rRNA* genes is a powerful approach when studying microbial community composition (*Caporaso et al., 2010*), but is limited in its ability to accurately discriminate very closely related bacteria. Sequences are clustered according to similarity but this is influenced by multiple factors including the length of the region sequenced, its evolutionary stability, and the error rate inherent to the sequencing chemistry used. Longer reads allow finer taxonomic discrimination, but higher error rates lead to larger numbers of spurious clusters, or operational taxonomic units (OTUs). The widely used clustering threshold of 97% identity (an approximation to species level clusters) can lead to closely related organisms being included in the same OTUs, as is the case with *B. longum* subspecies (*Bäckhed et al., 2015*). Quantitative PCR approaches to the discrimination of *B. longum* subspecies, utilizing *16S rRNA* gene primers, are problematic (*Youn, Seo & Ji, 2008*; *Tannock et al., 2013*) and whilst several molecular methods are currently able to discriminate bifidobacteria down to strain level (*Jarocki et al., 2016*) these are limited to culture isolates.

**Table 4 Relative abundances of *B. longum* subsp. *longum* and subsp. *infantis* in the feces of infants as detected by qPCR.**

| Geographical location | Nutrition | Delivery | Age at sampling (weeks) | *B. longum* subsp. *longum*[b] | *B. longum* subsp. *infantis*[b] |
|---|---|---|---|---|---|
| Chinese[a] | Breast milk | Caesarean | 6 | 16.6 (5.4) | 2.8 (1.9) |
| | | | 8 | 20.4 (6.6) | 4.4 (3.6) |
| | | | 10 | 26.4 (6.5) | 3.6 (3.6) |
| | | | 12 | 28.0 (7.3) | 1.4 (1.4) |
| | Cow's milk formula | Caesarean | 6 | 6.6 (3.0) | 1.7 (1.3) |
| | | | 8 | 11.3 (5.0) | 2.3 (1.7) |
| | | | 10 | 12.3 (4.6) | 4.0 (3.4) |
| | | | 12 | 17.7 (6.2) | 0.6 (0.3) |
| Australian[a] | Breast milk | Vaginal | 8 | 33.7 (7.5) | 1.1 (1.1) |
| | | Caesarean | 8 | 13.9 (13.4) | 2.0 (2.0) |
| | Cow's milk formula | Vaginal | 8 | 30.1 (7.6) | 0.0 (0.0)[c] |
| | | Caesarean | 8 | 13.5 (10.4) | 0.0 (0.0)[c] |
| | Goat's milk formula | Vaginal | 8 | 16.1 (4.5) | 0.0 (0.0)[c] |
| | | Caesarean | 8 | 7.2 (4.7) | 0.0 (0.0)[c] |
| South-East Asian[a] | Breast milk supplemented with cow's milk formula | Vaginal | <1 | 7.3 (3.0) | 0.0 (0.0)[c] |
| | | | 8 | 4.9 (2.2) | 4.6 (4.6) |
| | | | 16 | 9.1 (7.5) | 1.3 (1.3) |
| | | Caesarean | <1 | 0.1 (0.1) | 0.0 (0.0)[c] |
| | | | 8 | 15.1 (4.9) | 0.2 (0.2) |
| | | | 16 | 18.9 (6.1) | 0.0 (0.0)[c] |

**Notes.**

[a] Chinese babies, $n = 51$ caesarean delivery/breast milk, $n = 40$ caesarean delivery/formula; Australian babies, $n = 30$ breast milk (23 vaginal delivery, 7 caesarean), $n = 30$ cow's milk formula (21 vaginal delivery, 9 caesarean), $n = 30$ goat's milk formula (21 vaginal, 9 caesarean); South-East Asian babies, $n = 12$ vaginally delivery/breast milk and/or cow's milk formula, $n = 12$ caesarean delivery/breast milk and/or cow's milk formula.
[b] Mean % (SEM) of total microbiota determined by differential qPCR.
[c] Less than 0.01%.

As we demonstrate here, quantitative discrimination of bifidobacterial subspecies can be achieved using specific primers and qPCR probes that target unique functional gene sequences. Our data obtained from infants of different ethnicities and culture support the view that, of the two subspecies, *longum* predominates in infant feces (*Grönlund et al., 2011*; *Huda et al., 2014*; *Martin et al., 2016*). This might be unexpected, especially in the case of breast milk-fed infants where the specialized HMO metabolism of subspecies *infantis* could favor proliferation of this subspecies over subspecies *longum* (*Sela et al., 2008*). Investigations of the infant bowel microbiota in terms of comparative bacterial growth using available substrates and resource partitioning in the community are required to provide ecological explanations of bifidobacterial relative abundances.

The evolutionary importance of the infant-bifidobacteria paradigm remains speculative, but may involve competitive exclusion of pathogens and/or support of infant nutrition (*Kunz et al., 2000*; *D'Aimmo et al., 2012*; *Gordon et al., 2012*; *Ruhaak et al., 2014*). The resolution of this quandary requires an understanding of the roles of the different bifidobacterial species and subspecies in the infant bowel, how they form co-operative consortia, and

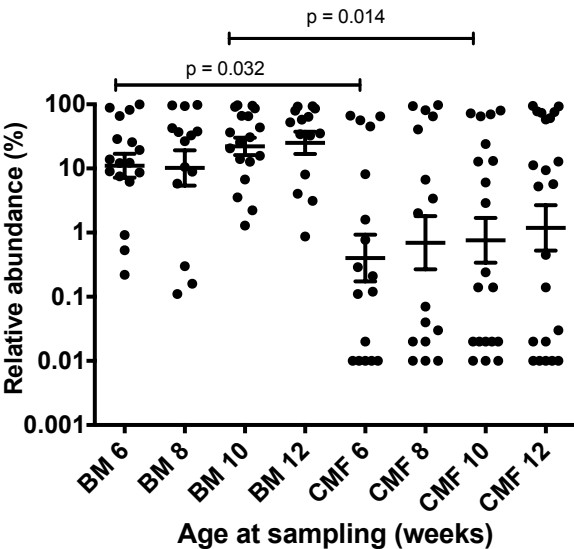

**Figure 1** **Effect of baby nutrition (breast milk, cow's milk formula) on relative abundances of *B. longum* subsp. *longum* in the feces of caesarean-delivered Chinese children.** The relative abundance (*B. longum* subsp. *longum* abundance/Total *16S rRNA* gene target abundance) data was normalized by log transformation and evaluated statistically by one-way ANOVA with multiple comparisons. Scatter plots with means and SEM are shown. Subspecies *longum* abundances were more variable in the feces of babies fed cow's milk formula (CMF), compared to those of breast milk-fed infants (BM) of the same ages. Statistically significant differences were observed between the six and 10 week sampling groups. The test limit of detection was 0.01% relative abundance.

how their activities impact on infant health. Knowledge of the relative proportions of the different kinds of bifidobacteria is necessary for such studies to advance. Using our method, subspecies *longum* and *infantis* can now be easily monitored in feces collected from infants born in different locations and in different human societies where cultural practices may influence bowel ecology in early life. This data may be important in developing interventions (such as the use of novel oligosaccharides) to promote bifidobacterial consortia in the bowel that influence infant development.

## CONCLUSION

Bifidobacteria, represented by several species, are numerous in the feces of infants during the exclusively milk-fed period of life. It is important to know the ecology of these species if health-promoting interventions of the bowel microbiota are to proceed. We describe the development and testing of a method by which two subspecies of *Bifidobacterium longum* can be differentiated quantitatively, according to strain-specific functional targets, in infant microbiota studies.

## ACKNOWLEDGEMENTS

We acknowledge Thomas Sullivan's contribution to the Australian study, and Jan Knol as Danone global platform director Gut Biology and Microbiology.

### Funding

The work was supported by MBIE contract UOOX1405. The funders had no role in study design, data collection and analysis, decision to publish, or preparation of the manuscript.

### Grant Disclosures

The following grant information was disclosed by the authors:
MBIE: UOOX1405.

### Competing Interests

The authors declare there are no competing interests. Colin G. Prosser and Dianne Lowry are employees of Dairy Goat Cooperative (NZ) Ltd., Hamilton, New Zealand. Pheng Soon Lee and Khai Hong Wong are employees of Mead Johnson Nutrition, Marina Bay, Singapore.

### Author Contributions

- Blair Lawley and Gerald W. Tannock conceived and designed the experiments, performed the experiments, analyzed the data, wrote the paper, prepared figures and/or tables, reviewed drafts of the paper.
- Karen Munro and Alan Hughes performed the experiments.
- Alison J. Hodgkinson, Colin G. Prosser, Dianne Lowry, Shao J. Zhou, Maria Makrides, Robert A. Gibson, Christophe Lay, Charmaine Chew and Pheng Soon Lee contributed reagents/materials/analysis tools, reviewed drafts of the paper.
- Khai Hong Wong reviewed drafts of the paper.

### Ethics

The following information was supplied relating to ethical approvals (i.e., approving body and any reference numbers):

For Australian infants: Ethical approval for the work contained in this article was obtained from the WCHN Human Research Ethics Committee. The infants were part of a larger study: Australian New Zealand Clinical Trials Registry ACTRN12608000047392.

For South-East Asian infants: The infants (various ethnicities) were part of a larger clinical study registered under the Dutch Trial Register (NTR 2838).

For Chinese infants: Infants were part of part of Mead Johnson Nutrition Clinical Protocol Number 8602. Ethical approval for the work contained in this article was obtained from the Ethics Committee, Xin Hua Hospital, affiliated to the School of Medicine, Shanghai Jiao Tong University, China.

### Data Availability

The raw data has been supplied as a Supplementary File.

## Supplemental Information

Supplemental information for this article can be found online at http://dx.doi.org/10.7717/peerj.3375#supplemental-information.

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
