# Peer review of "Differentiation of Bifidobacterium longum subspecies longum and infantis by quantitative PCR using functional gene targets"

_PeerJ, doi:10.7717/peerj.3375_

## Round 0.1 · original submission · Major Revisions

Please response to the reviewers comments especially the first reviewer in your revised manuscript.

Reviewer 1 ·

Basic reporting

"Differentiation of Bifidobacterium longum subspecies longum and infantis by quantitative PCR using functional gene targets" by Lawley et al presents a qPCR assay to differentiate B. infantis from B. longum, two gut microbes undifferentiable by their 16S sequence. They use functional genes to develop primers and probe, and later test their assay agains fecal DNA from infants from three countries. Technically there is little information to judge the assays, and the manuscript as presented is very preliminar. I suggest to rewrite the manuscript focusing more on the fecal genomic DNA than the assay.
The manuscript is not well written. Results are not fully presented or well described, the discussion is a two-paragraph reedition of some results, and the conclusion is meaningless.
References are ok, although it appears that journal names should be capitalized
Considering that this manuscript presents a qPCR assay which is later evaluated using fecal genomic DNA, I´m not sure if the clinical trials that generated those DNAs should be reported.
Discussion only restates a few results, where it should present a broader discussion of the importance of the work presented. Conclusion must be rewritten.

L23: change exclusively-milk-diet
L25: the other subspecies (suis) is also found in the bowel microbiota, but not human
L28 and throughout the MS: check Marchesi et al 2015 for proper usage (16 rRNA)
L36: "pangenomic search" might be an overstatement
L38: change total
L38: change "gold standard" strains, there are no gold standard strains
L53-57: there is no need to mention >10 works to indicate that bifidobacteria are dominant
L60: there is no mention to the third subspecies in the introduction. Later, the qPCR assays are not tested against DNA of this subspecies (suis). Authors should reference important work on this field, from Mattareli et al 2008 Int J Syst Evol Microbiol, and Karocki et al 2016 BMC Microbiology
L90: this is part of results
L214: in three different

Experimental design

The research question is not well defined, what is the real importance or need of having a qPCR assay differentiating B. infantis from B. longum.
The authors base their assays in the search of "functional differences" between B. longum subspecies. I could understand a sialidase being a target for B. infantis, but no mention is made if the respective gene is well conserved among all public genomes. Could this functional difference be present in all isolates? Moreover, the target chosen for B. longum is a "sugar kinase". No rationale is given for this selection, and how a sugar kinase could represent a functional difference present in all B. longum is not explained.
In any case, results for the selectivity of the assays are not presented. I suggest to present a sequence alignment, or a dendrogram of sialidases and the sugar kinase.
Moreover, more real evidence of the specificity of the assays should be done against other gut microbes, especially Bacteroides which are well known for sialidases and sugar kinases.
One flaw in this work is that qPCRs were done only in duplicates, while MIQE standards suggest qPCR assays should be in triplicate. No major information about qPCR standard deviations is presented.
In any case, the specificity of the assays is not shown (suggest using a Table or supp figure), and sensitivity of the assays is also an important aspect of the assays, how much they can detect.

L122: How incubations were done
Table 2: should include SD

Validity of the findings

It is hard to evaluate robustness since most qPCR data is not shown.
Conclusion is poorly presented, not linked to any question or results.

Reviewer 2 ·

Basic reporting

Clear and unambiguous, professional English used throughout.
- Good

Literature references, sufficient field background/context provided.
- The introductory provides concise background information.

Professional article structure, figs, tables. Raw data shared
- Article structure: Good
- Minor modifications were required in the tables.

Self-contained with relevant results to hypotheses.
- Good

Experimental design

Original primary research within Aims and Scope of the journal.
- Good.

Rigorous investigation performed to a high technical & ethical standard.
- Good.

Methods described with sufficient detail & information to replicate.
- Fair. Need minor modifications as mentioned in general comments.

Validity of the findings

The reviewer completely agree with the usefulness and robustness of the specific primers targeting functional gene exclusive to each of these subspecies.
I was surprised with the data showing that B. longum subspecies longum are more dominant bifodobacteria than subspecies infantis in breast-fed infants. It might be the difference between country or study design (days after birth, nutrition, etc).

Additional comments

The manuscript by Lawley and co-workers reports on the development of real-time PCR assay to discriminate Bifidobacterium longum subspecies longum and infantis. Discrimination of these subspecies would attract attention for researchers working on human (especially infant) gut microbiota. The authors developed specific PCR primers and TaqMan probes targeting functional gene exclusive to each of these subspecies. Although some specific primer for these subspecies have been reported, I think that there are some progress and conceptual advance.
The other appraisable point is the application of the specific-PCR method. They used DNA extracted from the feces of large number of infants from different countries (n=205). I believe that it is important to investigate the prevalence of these subspecies in larger number of samples to discuss the correlation with the difference of country, delivery mode, age, and nutrition; however it is beyond the aim of this study.
Overall, manuscript is well written and I evaluate the study provide a useful tool for analyzing human gut microbiota. I would like to suggest some points as you can see below. I hope that these comments would help you improving the manuscript.

Major point 1.
Figure 1 is difficult to understand. What does y-axis (log relative abundance) represent? Does it represents relative abundance of B. longum subspecies longum against subspecies infantis? Does it represents relative abundance (percentage?) of B. longum subspecies longum against total bacteria? If the later interpretation was the case, I would like to suggest adding the similar figure for B. infantis. This point have to be clearly described or improved before publication.

Minor point 1 (line 87 and 88).
Accession ID of the complete genome of B. longum subsp. longum strain ATCC 15697 should clearly be stated. I recommend the author to explain that Blon2348 is not the accession ID but “locus_tag”. Accession ID of the complete genome of strain NCC 2705 should be stated as well.

Minor point 2 (Table 1, 3, 4)
There are obstructive cross-bar in these tables and I recommend improving the table format. I would like to propose “improved Table 1”, which you can find as an attached Excel file. I would expect that Table 3 and 4 will be improved accordingly.

Annotated reviews are not available for download in order to protect the identity of reviewers who chose to remain anonymous.

---

## Round 0.2 · Minor Revisions

Please revise your manuscript and send it to us s as soon as possible

Reviewer 1 ·

Basic reporting

no comment

Experimental design

no comment

Validity of the findings

no comment

Additional comments

Most concerns I had in the previous version were addressed, which in part validate them. The importance of the study is better presented, and details regarding the assay developed are also indicated. It is important that the manuscript indicates what genes were chosen and why. Validity of the assays appears better explained in the methods and results section.

- I´m still not convinced that it is necessary to include 10 references to indicate the importance and dominance of bifidobacteria in the infant gut.
- Table 2: should include SD. This was not addressed by the authors.
- The results section needs to briefly restate the major goals of the study.
- Conclusion needs to better indicate what was done in the study (more than “describe a method”), and what conclusions were obtained in the infant fecal samples. This was not addressed by the authors.
- In the new Table added in this version, locus tags for B. longum should be “BL274”?
- The discussion should at least provide a comment on previous methods used to differentiate both subspecies, giving more context to this study.

Reviewer 2 ·

Basic reporting

As I commented on the 1st review report, basic reporting is good enough.

Experimental design

As I commented on the 1st review report, experimental design is good enough.
Methods described with sufficient detail & information to replicate has been improved according to the comments by reviewers.

Validity of the findings

I evaluate the study since the conclusion is well stated, linked to original research question and limited to supporting results. The tool is useful and the study have impact and novelty on the research field.

Additional comments

The revised manuscript by Lawley and co-workers has been improved according to the comments made by the reviewers. I (Reviewer 2) have additional comments on Figure 1.

I recommend modifying the label of y-axis as pointed below.
"Log relative abundance" to "Relative abundance (%)"
"3" to be deleted. "2" to be "100". "1" to be "10"."0" to be "1". "-1" to be "0.1". "-2" to be "0.01". "-3" to be deleted or "0.001".
Detection limit of the specific-PCR assay is 0.01% and plotted in this figure, isn't it? If it was the case, detection limit should be mentioned in the figure footnote.

> B. longum subsp. infantis abundance and prevalence was low in the cohort represented in figure 1.
> Addition of a similar figure for B. infantis would only highlight the relative lack of this subspecies
> rather than illuminate any trends across diet.

As you mentioned in introduction, many researchers are interested in the distribution of B. longum subspecies longum and infantis. So I believe that many of the potential readers (including me) want to know how does the B. longum subspecies infantis were detected along the week after birth in your cohort. Although summarized data are presented in Table 3, we don't see any inter-individual variation. So I would like to 'recommend' showing the data of B. longum subspecies infantis in Fig. 1 panel "B" or supplementary Fig.

---

## Round 0.3 · accepted · Accept

Congratulations , and thanks for your submission.